# Exponential ergodicity of mirror-Langevin diffusions

**Sinho Chewi**
MIT
schewi@mit.edu

**Thibaut Le Gouic**
MIT
tlegouic@mit.edu

**Chen Lu**
MIT
chenl819@mit.edu

**Tyler Maunu**
MIT
maunut@mit.edu

**Philippe Rigollet**
MIT
rigollet@mit.edu

**Austin Stromme**
MIT
astromme@mit.edu

## Abstract

Motivated by the problem of sampling from ill-conditioned log-concave distributions, we give a clean non-asymptotic convergence analysis of mirror-Langevin diffusions as introduced in [Zha+20]. As a special case of this framework, we propose a class of diffusions called Newton-Langevin diffusions and prove that they converge to stationarity exponentially fast with a rate which not only is dimension-free, but also has no dependence on the target distribution. We give an application of this result to the problem of sampling from the uniform distribution on a convex body using a strategy inspired by interior-point methods. Our general approach follows the recent trend of linking sampling and optimization and highlights the role of the chi-squared divergence. In particular, it yields new results on the convergence of the vanilla Langevin diffusion in Wasserstein distance.

## 1 Introduction

Sampling from a target distribution is a central task in statistics and machine learning with applications ranging from Bayesian inference [RC04; DM+19] to deep generative models [Goo+14]. Owing to a firm mathematical grounding in the theory of Markov processes [MT09], as well as its great versatility, Markov Chain Monte Carlo (MCMC) has emerged as a fundamental sampling paradigm. While traditional theoretical analyses are anchored in the asymptotic framework of ergodic theory, this work focuses on finite-time results that better witness the practical performance of MCMC for high-dimensional problems arising in machine learning.

This perspective parallels an earlier phenomenon in the much better understood field of optimization where convexity has played a preponderant role for both theoretical and methodological advances [Nes04; Bub15]. In fact, sampling and optimization share deep conceptual connections that have contributed to a renewed understanding of the theoretical properties of sampling algorithms [Dal17a; Wib18] building on the seminal work of Jordan, Kinderlehrer and Otto [JKO98].

We consider the following canonical sampling problem. Let $\pi$ be a log-concave probability measure over $\mathbb{R}^d$ so that $\pi$ has density equal to $e^{-V}$, where the potential $V : \mathbb{R}^d \to \mathbb{R}$ is convex. Throughout this paper, we also assume that $V$ is twice continuously differentiable for convenience, though many of our results hold under weaker conditions.

Most MCMC algorithms designed for this problem are based on the *Langevin diffusion* (LD), that is the solution $(X_t)_{t \geq 0}$ to the stochastic differential equation (SDE)

$$\mathrm{d}X_t = -\nabla V(X_t)\,\mathrm{d}t + \sqrt{2}\,\mathrm{d}B_t, \tag{LD}$$

with $(B_t)_{t\geq 0}$ a standard Brownian motion in $\mathbb{R}^d$. Indeed, $\pi$ is the unique invariant distribution of (LD) and suitable discretizations result in algorithms that can be implemented when $V$ is known only up to an additive constant, which is crucial for applications in Bayesian statistics and machine learning.

A first connection between sampling from log-concave measures and optimizing convex functions is easily seen from (LD): omitting the Brownian motion term yields the gradient flow $\dot{x}_t = -\nabla V(x_t)$, which results in the celebrated gradient descent algorithm when discretized in time [Dal17a; Dal17b]. There is, however, a much deeper connection involving the distribution of $X_t$ rather than $X_t$ itself, and this latter connection has been substantially more fruitful: the marginal distribution of a Langevin diffusion process $(X_t)_{t\geq 0}$ evolves according to a *gradient flow*, over the Wasserstein space of probability measures, that minimizes the Kullback-Leibler (KL) divergence $D_{\mathrm{KL}}(\cdot \parallel \pi)$ [JKO98; AGS08; Vil09]. This point of view has led not only to a better theoretical understanding of the Langevin diffusion [Ber18; CB18; Wib18; DMM19; VW19] but it has also inspired new sampling algorithms based on classical optimization algorithms, such as proximal/splitting methods [Ber18; Wib18; Wib19; SKL20], mirror descent [Hsi+18; Zha+20], Nesterov's accelerated gradient descent [Che+18; Ma+19; DR20], and Newton methods [Mar12; Sim+16; WL20].

**Our contributions.** This paper further exploits the optimization perspective on sampling by establishing a theoretical framework for a large class of stochastic processes called *mirror-Langevin diffusions* (MLD) introduced in [Zha+20]. These processes correspond to alternative optimization schemes that minimize the KL divergence over the Wasserstein space by changing its geometry. They show better dependence in key parameters such as the condition number and the dimension.

Our theoretical analysis is streamlined by a technical device which is unexpected at first glance, yet proves to be elegant and effective: we track the progress of these schemes not by measuring the objective function itself, the KL divergence, but rather by measuring the chi-squared divergence to the target distribution $\pi$ as a surrogate. This perspective highlights the central role of mirror Poincaré inequalities (MP) as sufficient conditions for exponentially fast convergence of the mirror-Langevin diffusion to stationarity in chi-squared divergence, which readily yields convergence in other well-known information divergences, such as the Kullback-Leibler divergence, the Hellinger distance, and the total variation distance [Tsy09, §2.4].

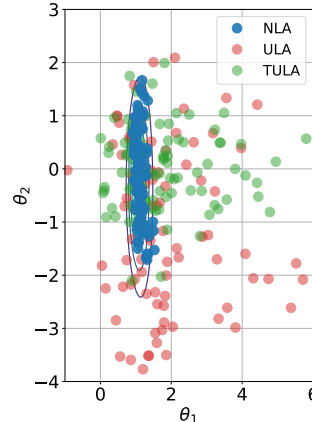

We also specialize our results to the case when the mirror map equals the potential $V$. This can be understood as the sampling analogue of Newton's method, and we therefore call it the *Newton-Langevin diffusion* (NLD). In this case, the mirror Poincaré inequality translates into the Brascamp-Lieb inequality which automatically holds when $V$ is twice-differentiable and *strictly* convex. In turn, it readily implies exponential convergence of the Newton-Langevin diffusion (Corollary 1) and can be used for approximate sampling even when the second derivative of $V$ vanishes (Corollary 2). Strikingly, the rate of convergence *has no dependence on $\pi$ or on the dimension $d$* and, in particular, is robust to cases where $\nabla^2 V$ is arbitrarily close to zero. This *scale-invariant* convergence parallels that of Newton's method in convex optimization and is the first result of this kind for sampling.

This invariance property is useful for approximately sampling from the uniform distribution over a convex body $\mathcal{C}$, which has been well-studied in the computer science literature [FKP94; KLS95; LV07]. By taking the target distribution $\pi \propto \exp(-\beta V)$, where $V$ is any strictly convex *barrier function*, and $\beta$, the inverse temperature parameter, is taken to be small (depending on the target accuracy), we can use the

Figure 1: Samples from the posterior distribution of a 2D Bayesian logistic regression model using the Newton-Langevin Algorithm (NLA), the Unadjusted Langevin Algorithm (ULA), and the Tamed Unadjusted Langevin Algorithm (TULA) [Bro+19]. For details, see Section E.2.

Newton-Langevin diffusion, much in the spirit of interior point methods (as promoted by [LTV20]), to output a sample which is approximately uniformly distributed on $\mathcal{C}$; see Corollary 3.

Throughout this paper, we work exclusively in the setting of continuous-time diffusions such as (LD). We refer to the works [DM15; Dal17a; Dal17b; RRT17; CB18; Wib18; DK19; DMM19; DRK19; Mou+19; VW19] for discretization error bounds, and leave this question open for future works.

**Related work.** The discretized Langevin algorithm, and the Metropolis-Hastings adjusted version, have been well-studied when used to sample from strongly log-concave distributions, or distributions satisfying a log-Sobolev inequality [Dal17b; DM17; CB18; Che+19; DK19; DM+19; Dwi+19; Mou+19; VW19]. Moreover, various ways of adapting Langevin diffusion to sample from bounded domains have been proposed [BEL18; Hsi+18; Zha+20]; in particular, [Zha+20] studied the discretized mirror-Langevin diffusion. Finally, we note that while our analysis and methods are inspired by the optimization perspective on sampling, it connects to a more traditional analysis based on coupling stochastic processes. Quantitative analysis of the continuous Langevin diffusion process associated to SDE (LD) has been performed with Poincaré and log-Sobolev inequalities [BGG12; BGL14; VW19], and with couplings of stochastic processes [CL89; Ebe16].

**Notation.** The Euclidean norm over $\mathbb{R}^d$ is denoted by $\|\cdot\|$. Throughout, we simply write $\int g$ to denote the integral with respect to the Lebesgue measure: $\int g(x)\,\mathrm{d}x$. When the integral is with respect to a different measure $\mu$, we explicitly write $\int g\,\mathrm{d}\mu$. The expectation and variance of $g(X)$ when $X \sim \mu$ are respectively denoted $\mathbb{E}_\mu\,g = \int g\,\mathrm{d}\mu$ and $\mathrm{var}_\mu\,g := \int (g - \mathbb{E}_\mu\,g)^2\,\mathrm{d}\mu$. When clear from context, we sometimes abuse notation by identifying a measure $\mu$ with its Lebesgue density.

## 2 Mirror-Langevin diffusions

Before introducing mirror-Langevin diffusions, our main objects of interest, we provide some intuition for their construction by drawing a parallel with convex optimization.

### 2.1 Gradient flows, mirror flows, and Newton's method

We briefly recall some background on gradient flows and mirror flows; we refer readers to the monograph [Bub15] for the convergence analysis of the corresponding discrete-time algorithms.

Suppose we want to minimize a differentiable function $f : \mathbb{R}^d \to \mathbb{R}$. The *gradient flow* of $f$ is the curve $(x_t)_{t\geq 0}$ on $\mathbb{R}^d$ solving $\dot{x}_t = -\nabla f(x_t)$. A suitable time discretization of this curve yields the well-known *gradient descent* (GD).

Although the gradient flow typically works well for optimization over Euclidean spaces, it may suffer from poor dimension scaling in more general cases such as Banach space optimization; a notable example is the case when $f$ is defined over the probability simplex equipped with the $\ell_1$ norm. This observation led Nemirovskii and Yudin [NJ79] to introduce the *mirror flow*, which is defined as follows. Let $\phi : \mathbb{R}^d \to \mathbb{R} \cup \{\infty\}$ be a *mirror map*, that is a strictly convex twice continuously differentiable function of *Legendre type*[1]. The mirror flow $(x_t)_{t\geq 0}$ satisfies $\partial_t \nabla\phi(x_t) = -\nabla f(x_t)$, or equivalently, $\dot{x}_t = -[\nabla^2\phi(x_t)]^{-1}\nabla f(x_t)$. The corresponding discrete-time algorithms, called *mirror descent* (MD) algorithms, have been successfully employed in varied tasks of machine learning [Bub15] and online optimization [BC12] where the entropic mirror map plays an important role. In this work, we are primarily concerned with the following choices for the mirror map:

1. When $\phi = \|\cdot\|^2/2$, then the mirror flow reduces to the gradient flow.

2. Taking $\phi = f$ and the discretization $x_{k+1} = x_k - h_k\,[\nabla^2 f(x_k)]^{-1}\nabla f(x_k)$ yields another popular optimization algorithm known as (damped) *Newton's method*. Newton's method has the important property of being invariant under affine transformations of the problem, and its local convergence is known to be much faster than that of GD; see [Bub15, §5.3].

### 2.2 Mirror-Langevin diffusions

We now introduce the *mirror-Langevin diffusion* (MLD) of [Zha+20]. Just as LD corresponds to the gradient flow, the MLD is the sampling analogue of the mirror flow. To describe it, let $\phi : \mathbb{R}^d \to \mathbb{R}$ be a mirror map as in the previous section. Then, the mirror-Langevin diffusion satisfies the SDE

$$X_t = \nabla\phi^\star(Y_t), \qquad \mathrm{d}Y_t = -\nabla V(X_t)\,\mathrm{d}t + \sqrt{2}\,[\nabla^2\phi(X_t)]^{1/2}\,\mathrm{d}B_t, \qquad \text{(MLD)}$$

where $\phi^\star$ denotes the convex conjugate of $\phi$ [BL06, §3.3]. In particular, if we choose the mirror map $\phi$ to equal the potential $V$, then we arrive at a sampling analogue of *Newton's method*, which we call the *Newton-Langevin diffusion* (NLD),

$$X_t = \nabla V^\star(Y_t), \qquad \mathrm{d}Y_t = -\nabla V(X_t)\,\mathrm{d}t + \sqrt{2}\,[\nabla^2 V(X_t)]^{1/2}\,\mathrm{d}B_t. \qquad \text{(NLD)}$$

From our intuition gained from optimization, we expect that NLD has special properties, such as affine invariance and faster convergence. We validate this intuition in Corollary 1 below by showing that, provided $\pi$ is strictly log-concave, the NLD converges to stationarity exponentially fast, with no dependence on $\pi$. This should be contrasted with the vanilla Langevin diffusion (LD), for which the convergence rate depends on the Poincaré constant of $\pi$, as we discuss in the next section.

We end this section by comparing MLD and NLD with similar sampling algorithms proposed in the literature inspired by mirror descent and Newton's method.

*Mirrored Langevin dynamics.* A variant of MLD, called "mirrored Langevin dynamics", was introduced in [Hsi+18]. The mirrored Langevin dynamics is motivated by constrained sampling and corresponds to the vanilla Langevin algorithm applied to the new target measure $(\nabla\phi)_\#\pi$. In contrast, MLD can be understood as a Riemannian diffusion w.r.t. the Riemannian metric induced by the mirror map $\phi$. Thus, the motivations and properties of the two algorithms are different, and we refer to [Zha+20] for further comparison of the two algorithms.

An earlier draft of [Hsi+18] also introduced MLD, along with a continuous-time analysis of the diffusion. Their convergence analysis is based on the classical Bakry-Émery criterion (see [BGL14]), which is generally harder to check than the mirror Poincaré inequality (MP) that we introduce below; in particular, when $\phi = V$, we show that the mirror Poincaré inequality holds automatically.

*Quasi-Newton diffusion.* The paper [Sim+16] proposes a quasi-Newton sampling algorithm, based on L-BFGS, which is partly motivated by the desire to avoid computation of the third derivative $\nabla^3 V$ while implementing the Newton-Langevin diffusion. We remark, however, that the form of NLD employed above, which treats $V$ as a mirror map, does not in fact require the computation of $\nabla^3 V$, and thus can be implemented practically; see Section 5. Moreover, since we analyze the full NLD, rather than a quasi-Newton implementation, we are able to give a clean convergence result.

*Information Newton's flow.* Inspired by the perspective of [JKO98], which views the Langevin diffusion as a gradient flow in the Wasserstein space of probability measures, the paper [WL20] proposes an approach termed "information Newton's flow" that applies Newton's method directly on the space of probability measures equipped with either the Fisher-Rao or the Wasserstein metric. However, unlike LD and NLD that both operate at the level of particles, information Newton's flow faces significant challenges at the level of both implementation and analysis.

## 3 Convergence analysis

### 3.1 Convergence of gradient flows and mirror flows

We provide a brief reminder about the convergence analysis of gradient flows and mirror flows defined in Section 2.1 to provide intuition for the next section. Throughout, let $f$ be a differentiable function with minimizer $x^*$.

Consider first the gradient flow for $f$: $\dot{x}_t = -\nabla f(x_t)$. We get $\partial_t[f(x_t) - f(x^*)] = -\|\nabla f(x_t)\|^2$ from a straightforward computation. From this identity, it is natural to assume a *Polyak-Łojasiewicz* (PL) *inequality*, which is well-known in the optimization literature [KNS16] and can be employed even when $f$ is not convex [Che+20]. Indeed, if there exists a constant $C_{\mathsf{PL}} > 0$ with

$$f(x) - f(x^*) \leq \frac{C_{\mathsf{PL}}}{2}\,\|\nabla f(x)\|^2 \qquad \forall\,x \in \mathbb{R}^d\,, \qquad \text{(PL)}$$

then $\partial_t[f(x_t) - f(x^*)] \leq -\frac{2}{C_{\mathsf{PL}}}\,[f(x_t) - f(x^*)]$. Together with Grönwall's inequality, it readily yields exponentially fast convergence in objective value: $f(x_t) \leq f(x_0)\,e^{-2t/C_{\mathsf{PL}}}$.

A similar analysis may be carried out for the mirror flow. Fix a mirror map $\phi$ and consider the mirror flow: $\dot{x}_t = -[\nabla^2\phi(x_t)]^{-1}\nabla f(x_t)$. It holds $\partial_t[f(x_t) - f(x^*)] = -\langle\nabla f(x_t), [\nabla^2\phi(x_t)]^{-1}\nabla f(x_t)\rangle$.

Therefore, the analogue of (PL) which guarantees exponential decay in the objective value is the following inequality, which we call a *mirror PL inequality*:

$$f(x) - f(x^*) \leq \frac{C_{\mathsf{MPL}}}{2} \langle \nabla f(x), [\nabla^2 \phi(x)]^{-1} \nabla f(x) \rangle \qquad \forall x \in \mathbb{R}^d. \tag{MPL}$$

Next, we describe analogues of (PL) and (MPL) that guarantee convergence of LD and MLD.

### 3.2 Convergence of mirror-Langevin diffusions

The above analysis employs the objective function $f$ to measure the progress of both the gradient and mirror flows. While this is the most natural choice, our approach below crucially relies on measuring progress via a *different functional $F$*. What should we use as $F$? To answer this question, we first consider the simpler case of the vanilla Langevin diffusion (LD), which is a special case of MLD when the mirror map is $\phi = \|\cdot\|^2/2$. We keep this discussion informal and postpone rigorous arguments to Appendix A.

Since the work of [JKO98], it has been known that the marginal distribution $\mu_t$ at time $t \geq 0$ of LD evolves according to the *gradient flow* of the KL divergence $D_{\mathrm{KL}}(\cdot \parallel \pi)$ with respect to the 2-Wasserstein distance $W_2$; we refer readers to [San17] for an overview of this work, and to [AGS08; Vil09] for comprehensive treatments. Therefore, the most natural choice for $F$ is, as in Section 3.1, the objective function $D_{\mathrm{KL}}(\cdot \parallel \pi)$ itself. Following this approach, one can compute [Vil03, §9.1.5]

$$\partial_t D_{\mathrm{KL}}(\mu_t \parallel \pi) = -\int \left\| \nabla \ln \frac{\mathrm{d}\mu_t}{\mathrm{d}\pi} \right\|^2 \mathrm{d}\mu_t = -4 \int \left\| \nabla \sqrt{\frac{\mathrm{d}\mu_t}{\mathrm{d}\pi}} \right\|^2 \mathrm{d}\pi.$$

In this setup, the role of the PL inequality (PL) is played by a *log-Sobolev inequality* of the form

$$\mathrm{ent}_\pi(g^2) := \int g^2 \ln(g^2) \, \mathrm{d}\pi - \left( \int g^2 \, \mathrm{d}\pi \right) \ln \left( \int g^2 \, \mathrm{d}\pi \right) \leq 2C_{\mathsf{LSI}} \int \|\nabla g\|^2 \, \mathrm{d}\pi. \tag{LSI}$$

When $g = \sqrt{\mathrm{d}\mu_t/\mathrm{d}\pi}$, (LSI) reads $D_{\mathrm{KL}}(\mu_t \parallel \pi) \leq 2C_{\mathsf{LSI}} \int \left\| \nabla \sqrt{\mathrm{d}\mu_t/\mathrm{d}\pi} \right\|^2 \mathrm{d}\pi$, which implies exponentially fast convergence: $D_{\mathrm{KL}}(\mu_t \parallel \pi) \leq D_{\mathrm{KL}}(\mu_0 \parallel \pi) \, e^{-2t/C_{\mathsf{LSI}}}$ by Grönwall's inequality.

A disadvantage of this approach, however, is that the log-Sobolev inequality (LSI) does not hold for any log-concave measure $\pi$, or it may hold with a poor constant $C_{\mathsf{LSI}}$. For example, it is known that the log-Sobolev constant of an isotropic log-concave distribution must in general depend on the diameter of its support [LV18]. In contrast, we work below with a *Poincaré inequality*, which is conjecturally satisfied by such distributions with a *universal constant* [KLS95].

Motivated by [BCG08; CG09], we instead consider the *chi-squared divergence*

$$F(\mu) = \chi^2(\mu \parallel \pi) := \mathrm{var}_\pi \frac{\mathrm{d}\mu}{\mathrm{d}\pi} = \int \left( \frac{\mathrm{d}\mu}{\mathrm{d}\pi} \right)^2 \mathrm{d}\pi - 1, \qquad \text{if } \mu \ll \pi,$$

and $F(\mu) = \infty$ otherwise. It is well-known that the law $(\mu_t)_{t \geq 0}$ of LD satisfies the Fokker-Planck equation in the weak sense [KS91, §5.7]:

$$\partial_t \mu_t = \mathrm{div}\big( \mu_t \, \nabla \ln \frac{\mu_t}{\pi} \big).$$

Using this, we can compute the derivative of the chi-squared divergence:

$$\frac{1}{2} \partial_t F(\mu_t) = \int \frac{\mu_t}{\pi} \, \partial_t \mu_t = \int \frac{\mu_t}{\pi} \, \mathrm{div}\big( \mu_t \nabla \ln \frac{\mu_t}{\pi} \big) = -\int \langle \nabla \ln \frac{\mu_t}{\pi}, \nabla \frac{\mu_t}{\pi} \rangle \mu_t = -\int \left\| \nabla \frac{\mu_t}{\pi} \right\|^2 \pi,$$

and exponential convergence of the chi-squared divergence follows if $\pi$ satisfies a Poincaré inequality:

$$\mathrm{var}_\pi g \leq C_{\mathsf{P}} \, \mathbb{E}_\pi[\|\nabla g\|^2] \qquad \text{for all locally Lipschitz } g \in L^2(\pi). \tag{P}$$

Thus, when using the chi-squared divergence to track progress, the role of the PL inequality is played by a Poincaré inequality. As we discuss in Sections 4.1 and 4.3 below, the Poincaré inequality is significantly weaker than the log-Sobolev inequality.

A similar analysis may be carried out for MLD using an appropriate variation of Poincaré inequalities.

**Definition 1** (Mirror Poincaré inequality)**.** Given a mirror map $\phi$, we say that the distribution $\pi$ satisfies a *mirror Poincaré inequality* with constant $C_{\mathsf{MP}}$ if

$$\operatorname{var}_\pi g \leq C_{\mathsf{MP}} \, \mathbb{E}_\pi \langle \nabla g, (\nabla^2 \phi)^{-1} \nabla g \rangle \qquad \text{for all locally Lipschitz } g \in L^2(\pi). \qquad \text{(MP)}$$

When $\phi = \|\cdot\|^2/2$, (MP) is simply called a *Poincaré inequality* and the smallest $C_{\mathsf{MP}}$ for which the inequality holds is the *Poincaré constant* of $\pi$, denoted $C_{\mathsf{P}}$.

Using a similar argument as the one above, we show exponential convergence of MLD in $\chi^2(\cdot \parallel \pi)$ under (MP). Together with standard comparison inequalities between information divergences [Tsy09, §2.4], it implies exponential convergence in a variety of commonly used divergences, including the total variation (TV) distance $\|\cdot - \pi\|_{\mathrm{TV}}$, the Hellinger distance $H(\cdot, \pi)$, and the KL divergence $D_{\mathrm{KL}}(\cdot \parallel \pi)$.

**Theorem 1.** *For each $t \geq 0$, let $\mu_t$ be the marginal distribution of* MLD *with target distribution $\pi$ at time $t$. Then if $\pi$ satisfies the mirror Poincaré inequality* (MP) *with constant $C_{\mathsf{MP}}$, it holds*

$$2\|\mu_t - \pi\|_{\mathrm{TV}}^2, \ H^2(\mu_t, \pi), \ D_{\mathrm{KL}}(\mu_t \parallel \pi), \ \chi^2(\mu_t \parallel \pi) \leq e^{-\frac{2t}{C_{\mathsf{MP}}}} \chi^2(\mu_0 \parallel \pi), \quad \forall t \geq 0 \, .$$

We give two proofs of this result in Appendix A.

Recall that LD can be understood as a gradient flow for the KL divergence on the 2-Wasserstein space. In light of this interpretation, the above bound for the KL divergence yields a convergence rate *in objective value*, and it is natural to wonder whether a similar rate holds for the iterates themselves in terms of 2-Wasserstein distance. From the works [Din15; Led18; Liu20], it is known that a Poincaré inequality (P) implies the transportation-cost inequality

$$W_2^2(\mu, \pi) \leq 2C_{\mathsf{P}} \chi^2(\mu \parallel \pi), \qquad \forall \mu \ll \pi. \qquad (1)$$

Initially unaware of these works, we proved that a Poincaré inequality implies a suboptimal chi-squared transportation inequality. Since the suboptimal inequality already suffices for our purposes, we state and prove it in Appendix B. We thank Jon Niles-Weed for bringing this to our attention.

The inequality (1) implies that if $\pi$ has a finite Poincaré constant $C_{\mathsf{P}}$ then Theorem 1 also yields exponential convergence in Wasserstein distance. In the rest of the paper, we write this result as

$$\frac{1}{2C_{\mathsf{P}}} W_2^2(\mu_t, \pi) \leq e^{-\frac{2t}{C_{\mathsf{MP}}}} \chi^2(\mu_0 \parallel \pi) \, ,$$

for *any* target measure $\pi$ that satisfies a mirror Poincaré inequality, with the convention that $C_{\mathsf{P}} = \infty$ when $\pi$ fails to satisfy a Poincaré inequality. In this case, the above inequality is simply vacuous.

## 4 Applications

We specialize Theorem 1 to the following important applications.

### 4.1 Newton-Langevin diffusion

For NLD, we assume that $V$ is strictly convex and twice continuously differentiable; take $\phi = V$. In this case, the mirror Poincaré inequality (MP) reduces to the *Brascamp-Lieb inequality*, which is known to hold with constant $C_{\mathsf{MP}} = 1$ for any strictly log-concave distribution $\pi$ [BL76; BL00; Gen08]. It yields the following remarkable result where the exponential contraction rate has no dependence on $\pi$ nor on the dimension $d$.

**Corollary 1.** *Suppose that $V$ is strictly convex and twice continuously differentiable. Then, the law $(\mu_t)_{t \geq 0}$ of* NLD *satisfies*

$$2\|\mu_t - \pi\|_{\mathrm{TV}}^2, \ H^2(\mu_t, \pi), \ D_{\mathrm{KL}}(\mu_t \parallel \pi), \ \chi^2(\mu_t \parallel \mu), \ \frac{1}{2C_{\mathsf{P}}} W_2^2(\mu_t, \pi) \leq e^{-2t} \chi^2(\mu_0 \parallel \pi).$$

If $\pi$ is log-concave, then it satisfies a Poincaré inequality [AB15; LV17] so that the result in Wasserstein distance holds. In fact, contingent on the famous *Kannan-Lovász-Simonovitz* (KLS) conjecture ([KLS95]), the Poincaré constant of any log-concave distribution $\pi$ is upper bounded by a constant, independent of the dimension, times the largest eigenvalue of the covariance matrix of $\pi$.

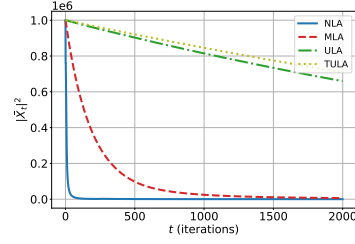

Figure 2: Approximately sampling from $\pi \propto e^{-\|\cdot\|}$ by sampling from $\pi_\beta \propto e^{-\|\cdot\|-\beta\|\cdot-\mathbf{1}\|^2}$ ($\beta = .0005$). Algorithms are initialized at a random $X_0$ with $\|X_0\| = 1000$. The plot shows the squared distance of the running means to 0.

At this point, one may wonder, under the same assumptions as the Brascamp-Lieb inequality, whether a mirror version of the log-Sobolev inequality (LSI) holds. This question was answered negatively in [BL00], thus reinforcing our use of the chi-squared divergence as a surrogate for the KL divergence.

If the potential $V$ is convex, but degenerate (i.e., not strictly convex) we cannot use NLD directly with $\pi$ as the target distribution. Instead, we perturb $\pi$ slightly to a new measure $\pi_\beta$, which is strongly log-concave, and for which we can use NLD. Crucially, due to the scale invariance of NLD, the time it takes for NLD to mix does not depend on $\beta$, the parameter which governs the approximation error.

**Corollary 2.** *Fix a target accuracy $\varepsilon > 0$. Suppose $\pi = e^{-V}$ is log-concave and set $\pi_\beta \propto e^{-V-\beta\|\cdot\|^2}$, where $\beta \leq \varepsilon^2/(2\int \|\cdot\|^2\,\mathrm{d}\pi)$. Then, the law $(\mu_t)_{t\geq 0}$ of NLD with target distribution $\pi_\beta$ satisfies $\|\mu_t - \pi\|_{\mathrm{TV}} \leq \varepsilon$ by time $t = \frac{1}{2}\ln[2\chi^2(\mu_0 \| \pi_\beta)] + \ln(1/\varepsilon)$.*

*Proof.* From our assumption, it holds

$$D_{\mathrm{KL}}(\pi \| \pi_\beta) = \int \ln\frac{\mathrm{d}\pi}{\mathrm{d}\pi_\beta}\,\mathrm{d}\pi = \beta\int \|\cdot\|^2\,\mathrm{d}\pi + \ln\int e^{-\beta\|\cdot\|^2}\,\mathrm{d}\pi \leq \beta\int \|\cdot\|^2\,\mathrm{d}\pi \leq \frac{\varepsilon^2}{2}.$$

Moreover, Theorem 1 with the above choice of $t$ yields $D_{\mathrm{KL}}(\mu_t \| \pi_\beta) \leq \varepsilon^2/2$. To conclude, we use Pinsker's inequality and the triangle inequality for $\|\cdot\|_{\mathrm{TV}}$. $\square$

Convergence guarantees for other cases where $\phi$ is only a *proxy* for $V$ are presented in Appendix C.

## 4.2 Sampling from the uniform distribution on a convex body

Next, we consider an application of NLD to the problem of sampling from the uniform distribution $\pi$ on a convex body $\mathcal{C}$. A natural method of outputting an approximate sample from $\pi$ is to take a strictly convex function $\widetilde{V} : \mathbb{R}^d \to \mathbb{R} \cup \{\infty\}$ such that $\mathrm{dom}\,\widetilde{V} = \mathcal{C}$ and $\widetilde{V}(x) \to \infty$ as $x \to \partial\mathcal{C}$, and to run NLD with target distribution $\pi_\beta \propto e^{-\beta\widetilde{V}}$, where the inverse temperature $\beta$ is taken to be small (so that $\pi_\beta \approx \pi$). The function $\widetilde{V}$ is known as a *barrier function*.

Although we can take any choice of barrier function $\widetilde{V}$, we obtain a clean theoretical result if we assume that $\widetilde{V}$ is $\nu^{-1}$-exp-concave, that is, the mapping $\exp(-\nu^{-1}\widetilde{V})$ is concave. Interestingly, this assumption further deepens the rich analogy between sampling and optimization, since such barriers are widely studied in the optimization literature. There, the property of exp-concavity is typically paired with the property of *self-concordance*, and barrier functions satisfying these two properties are a cornerstone of the theory of *interior point algorithms* (see [Bub15, §5.3] and [Nes04, §4]).

We now formulate our sampling result. In our continuous framework, it does not require self-concordance of the barrier function.

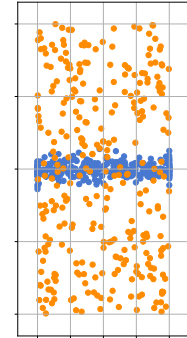

Figure 3: Uniform sampling from the set $[-0.01, 0.01] \times [-1, 1]$: PLA (blue) vs. NLA (orange). See Section E.3.

**Corollary 3.** *Fix a target accuracy $\varepsilon > 0$. Let $\pi$ be the uniform distribution over a convex body $\mathcal{C}$ and let $\widetilde{V}$ be a $\nu^{-1}$-exp-concave barrier for $\mathcal{C}$. Then, the law $(\mu_t)_{t\geq 0}$ of NLD with target density $\pi_\beta \propto e^{-\beta\widetilde{V}}$ for $\beta \leq \varepsilon^2/(2\nu)$ satisfies $\|\mu_t - \pi\|_{\mathrm{TV}} \leq \varepsilon$ by time $t = \frac{1}{2}\ln[2\chi^2(\mu_0 \| \pi_\beta)] + \ln(1/\varepsilon)$.*

*Proof.* Lemma 1 in Appendix D ensures that $D_{\mathrm{KL}}(\pi_\beta \,\|\, \pi) \leq \varepsilon^2/2$. We conclude as in the proof of Corollary 2, by using Theorem 1, Pinsker's inequality, and the triangle inequality for $\|\cdot\|_{\mathrm{TV}}$. $\qquad\square$

We demonstrate the efficacy of NLD in a simple simulation: sampling uniformly from the ill-conditioned rectangle $[-a, a] \times [-1, 1]$ with $a = 0.01$ (Figure 3). We compare NLA with the Projected Langevin Algorithm (PLA) [BEL18], both with 200 iterations and $h = 10^{-4}$. For NLA, we take $\widetilde{V}(x) = -\log(1 - x_1^2) - \log(a^2 - x_2^2)$ and $\beta = 10^{-4}$.

### 4.3  Langevin diffusion under a Poincaré inequality

We conclude this section by giving some implications of Theorem 1 to the classical Langevin diffusion (LD) when $\phi = \|\cdot\|^2/2$. In this case, the mirror Poincaré inequality (MP) reduces to the classical Poincaré inequality (P) as in Section 3.2.

**Corollary 4.** *Suppose that $\pi$ satisfies a Poincaré inequality* (P) *with constant $C_{\mathsf{P}} > 0$. Then, the law $(\mu_t)_{t\geq 0}$ of the Langevin diffusion* (LD) *satisfies*

$$2\|\mu_t - \pi\|_{\mathrm{TV}}^2, \; H^2(\mu_t, \pi), \; D_{\mathrm{KL}}(\mu_t \,\|\, \pi), \; \chi^2(\mu_t \,\|\, \mu), \; \frac{1}{2C_{\mathsf{P}}} W_2^2(\mu_t, \pi) \leq e^{-\frac{2t}{C_{\mathsf{P}}}} \chi^2(\mu_0 \,\|\, \pi).$$

The convergence in TV distance recovers results of [Dal17b; DM17]. Bounds for the stronger error metric $\chi^2(\cdot \,\|\, \pi)$ have appeared explicitly in [CLL19; VW19] and is implicit in the work of [BCG08; CG09] on which the TV bound of [DM17] is based.

Moreover, it is classical that if $\pi$ satisfies a log-Sobolev inequality (LSI) with constant $C_{\mathsf{LSI}}$ then it has Poincaré constant $C_{\mathsf{P}} \leq C_{\mathsf{LSI}}$. Thus, the choice of the chi-squared divergence as a surrogate for the KL divergence when tracking progress indeed requires weaker assumptions on $\pi$.

## 5  Numerical experiments

In this section, we examine the numerical performance of the *Newton-Langevin Algorithm* (NLA), which is given by the following Euler discretization of NLD:

$$\nabla V(X_{k+1}) = (1 - h)\nabla V(X_k) + \sqrt{2h}\,[\nabla^2 V(X_k)]^{1/2}\xi_k, \tag{NLA}$$

where $(\xi_k)_{k\in\mathbb{N}}$ is a sequence of i.i.d. $\mathcal{N}(0, I_d)$ variables. In cases where $\nabla V$ does not have a closed-form inverse, such as the logistic regression case of Section E.2, we invert it numerically by solving the convex optimization problem $\nabla V^\star(y) = \operatorname{argmax}_{x\in\mathbb{R}^d} \{\langle x, y \rangle - V(x)\}$.

We focus here on sampling from an ill-conditioned generalized Gaussian distribution on $\mathbb{R}^{100}$ with $V(x) = \langle x, \Sigma^{-1} x \rangle^\gamma / 2$ for $\gamma = 3/4$ to demonstrate the scale invariance of NLD established in Corollary 1. Additional experiments, including the Gaussian case $\gamma = 1$, are given in Appendix E.

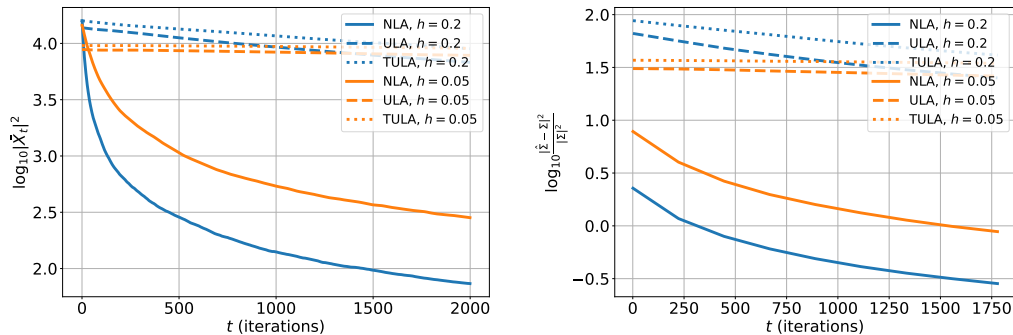

Figure 4: $V(x) = \langle x, \Sigma^{-1} x \rangle^{3/4}/2$, $\Sigma = \operatorname{diag}(1, 2, \ldots, 100)$. Left: absolute squared error of the mean 0. Right: relative squared error for the scatter matrix $\Sigma$.

Figure 4 compares the performance of NLA to that of the Unadjusted Langevin Algorithm (ULA) [DM+19] and of the Tamed Unadjusted Langevin Algorithm (TULA) [Bro+19]. We run the

algorithms 50 times and compute running estimates for the mean and scatter matrix of the family following [ZWG13]. Convergence is measured in terms of squared distance between means and relative squared distance between scatter matrices, $\|\hat{\Sigma} - \Sigma\|^2/\|\Sigma\|^2$. NLA generates samples that rapidly approximate the true distribution and also displays stability to the choice of the step size $h$.

## 6    Open questions

We conclude this paper by discussing several intriguing directions for future research. In this paper, we focused on giving clean convergence results for the continuous-time diffusions MLD and NLD, and we leave open the problem of obtaining discretization error bounds. In discrete time, Newton's method can be unstable, and one uses methods such as damped Newton, Levenburg-Marquardt, or cubic-regularized Newton [CGT00; NP06]; it is an interesting question to develop sampling analogues of these optimization methods. In a different direction, we ask the following question: are there appropriate variants of other popular sampling methods, such as accelerated Langevin [Ma+19] or Hamiltonian Monte Carlo [Nea12], which also enjoy the scale invariance of NLD?

### Broader impact

The sampling algorithms designed in this paper have the potential to improve a wide variety of Bayesian methods and therefore have an indirect impact on various domains such as health and medicine where such methods are pervasive. Sampling algorithms are also used for the generation of automated spam messages, which have potentially negative effects on society. Since this paper is primarily focused on theory, these questions are not addressed here.

### Acknowledgments

Philippe Rigollet was supported by NSF awards IIS-1838071, DMS-1712596, DMS-TRIPODS-1740751. Sinho Chewi and Austin Stromme were supported by the Department of Defense (DoD) through the National Defense Science & Engineering Graduate Fellowship (NDSEG) Program. Thibaut Le Gouic was supported by ONR grant N00014-17-1-2147 and NSF IIS-1838071.

We thank the reviewers for very helpful suggestions regarding the presentation of the paper.

## Footnotes

[1]This ensures that $\nabla\phi$ is invertible, c.f. [Roc97, §26].

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
