[Supplementary Material]

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

# A Proof of Theorem 1

The law $(\mu_t)_{t \geq 0}$ of MLD satisfies the Fokker-Planck equation

$$\partial_t \mu_t = \operatorname{div}\big(\mu_t \, (\nabla^2 \phi)^{-1} \nabla \ln \frac{\mu_t}{\pi}\big). \tag{2}$$

A unique solution to this equation, with enough regularity to justify our computations below, exists under fairly benign conditions on $\phi$ and $V$, see [LL08, Proposition 6].

As discussed in Section 3.2, it suffices to prove the convergence result in chi-squared divergence. The convergence results for total variation distance, Hellinger distance, and KL divergence follow from the inequalities [Tsy09, §2.4]

$$2\|\mu - \pi\|_{\mathrm{TV}}^2, \ H^2(\mu, \pi), \ D_{\mathrm{KL}}(\mu \parallel \pi) \leq \chi^2(\mu \parallel \pi), \qquad \forall \mu \ll \pi,$$

while the convergence in Wasserstein distance follows from (1).

*Proof of Theorem 1.* Using the Fokker-Planck equation (2), we may compute

$$\partial_t \chi^2(\mu_t \parallel \pi) = \partial_t \int \frac{\mu_t^2}{\pi} = 2 \int \frac{\mu_t}{\pi} \, \partial_t \mu_t = 2 \int \frac{\mu_t}{\pi} \, \operatorname{div}\big(\mu_t \, (\nabla^2 \phi)^{-1} \nabla \ln \frac{\mu_t}{\pi}\big)$$
$$= -2 \int \big\langle \nabla \frac{\mu_t}{\pi}, (\nabla^2 \phi)^{-1} \nabla \ln \frac{\mu_t}{\pi} \big\rangle \mu_t = -2 \int \big\langle \nabla \frac{\mu_t}{\pi}, (\nabla^2 \phi)^{-1} \nabla \frac{\mu_t}{\pi} \big\rangle \pi.$$

The mirror Poincaré inequality (MP) implies that this quantity is at most $-2 C_{\mathsf{MP}}^{-1} \chi^2(\mu_t \parallel \pi)$, which completes the proof via Grönwall's inequality. $\qquad \square$

We may reinterpret this proof within Markov semigroup theory.

*Proof of Theorem 1 from a Markov semigroup perspective.* We denote by $(P_t)_{t \geq 0}$ the semigroup of MLD; we refer readers to [BGL14; Han16] for background on Markov semigroup theory. The Dirichlet form $\mathcal{E}$ is given by

$$\mathcal{E}(f, g) = \int \langle \nabla f, (\nabla^2 \phi)^{-1} \nabla g \rangle \, \mathrm{d}\pi.$$

Since it is a self-adjoint semigroup, we get for all $f \in L^2(\pi)$,

$$\int P_t\big(\frac{\mathrm{d}\mu_0}{\mathrm{d}\pi}\big) f \, \mathrm{d}\pi = \int \big(\frac{\mathrm{d}\mu_0}{\mathrm{d}\pi}\big) P_t f \, \mathrm{d}\pi = \int P_t f \, \mathrm{d}\mu_0 = \int f \, \mathrm{d}\mu_t = \int \frac{\mathrm{d}\mu_t}{\mathrm{d}\pi} f \, \mathrm{d}\pi \, ,$$

so that

$$P_t\big(\frac{\mu_0}{\pi}\big) = \frac{\mu_t}{\pi}.$$

Therefore,

$$\chi^2(\mu_t \parallel \pi) := \mathrm{var}_\pi\Big(\frac{\mathrm{d}\mu_t}{\mathrm{d}\pi}\Big) = \mathrm{var}_\pi P_t\Big(\frac{\mathrm{d}\mu_0}{\mathrm{d}\pi}\Big).$$

Using a classical result of Markov semigroup theory (see for instance [CG09, Theorem 2.1] or [BGL14, Theorem 4.2.5])

$$\chi^2(\mu_t \parallel \pi) = \mathrm{var}_\pi P_t\Big(\frac{\mathrm{d}\mu_0}{\mathrm{d}\pi}\Big) \le e^{-\frac{2t}{C}} \, \mathrm{var}_\pi\Big(\frac{\mathrm{d}\mu_0}{\mathrm{d}\pi}\Big) = e^{-\frac{2t}{C}} \chi^2(\mu_0 \parallel \pi)$$

if and only if the semigroup $(P_t)_{t \ge 0}$ satisfies

$$\mathrm{var}_\pi(f) \le C\mathcal{E}(g, g), \qquad \text{for all } g \in D(\mathcal{E}), \tag{3}$$

where $\mathcal{E}$ is the Dirichlet form of $(P_t)_{t \ge 0}$ with domain $D(\mathcal{E})$. To conclude the proof, it suffices to note that (3) is precisely our assumption (MP) with $C = C_{\mathsf{MP}}$. $\qquad\square$

## B Convergence in 2-Wasserstein distance

### B.1 Background

As we have discussed, the proof of Theorem 1 in Appendix A implies that for any strictly log-concave target measure, the Newton-Langevin diffusion converges exponentially fast in the following error metrics: chi-squared divergence, KL divergence, Hellinger distance, and total variation distance. We also remark that convergence in Rényi divergences can also be proved in this setting, as in [VW19]. On the other hand, we would also like to know if we can obtain convergence results for *optimal transport* distances [Vil03]. As a first step, the transportation inequality of [Cor17],

$$D_{\mathrm{KL}}(\mu \parallel \pi) \ge \mathcal{T}_{D_V}(\mu \parallel \pi) := \inf\{\mathbb{E}\, D_V(X \parallel Z) : (X, Z) \text{ is a coupling of } (\mu, \pi)\},$$

which holds for all $\mu \ll \pi$, implies exponentially fast convergence in the asymmetric transportation cost $\mathcal{T}_{D_V}$, where $D_V(\cdot \parallel \cdot)$ is the Bregman divergence associated with $V$.

We turn towards the question of convergence in the 2-Wasserstein distance (denoted $W_2$). When $\pi$ is strongly log-concave, there is an elegant direct proof of exponential contraction in $W_2$ via a coupling of the Langevin process (see [Vil03, Exercise 9.10]). In general, however, convergence in $W_2$ is typically deduced from convergence in KL divergence, with the help of a *transportation-cost inequality*

$$W_2^2(\mu, \pi) \le C D_{\mathrm{KL}}(\mu \parallel \pi). \tag{4}$$

It has been known since the work of [OV00] that a log-Sobolev inequality (LSI) with constant $C_{\mathsf{LSI}}$ implies the validity of (4) with constant $C = C_{\mathsf{LSI}}$. Since an LSI may not always hold or may hold with a poor constant, [BV05] provides weaker conditions: namely, if there exists $\alpha > 0$ such that

$$\int \exp(\alpha\|x - x_0\|^2)\,\mathrm{d}\pi(x) < \infty, \tag{5}$$

then we have the weaker inequality

$$W_2^2(\mu, \pi) \lesssim D_{\mathrm{KL}}(\mu \parallel \pi) + \sqrt{D_{\mathrm{KL}}(\mu \parallel \pi)}.$$

Therefore, either the validity of an LSI or a square exponential moment suffice to transfer convergence in KL divergence to convergence in $W_2$. In fact, it turns out that the log-Sobolev inequality (LSI), the transportation inequality (4), and the square exponential moment condition (5) are all equivalent for log-concave measures, and they are in general strictly stronger than the Poincaré inequality (P) [Bob99; OV00; BV05].

Since Theorem 1 provides a stronger control, namely in chi-squared divergence rather than in KL divergence, the reader might wonder if a weaker transportation inequality in which the RHS of (4) is replaced by $C\chi^2(\mu \parallel \pi)^{1/p}$ might hold under weaker assumptions. Indeed, the recent works [Din15; Led18; Liu20] answer this question positively by showing that the Poincaré inequality (P) implies the transportation-cost inequality

$$W_2^2(\mu, \nu) \le C \inf_{p \ge 1}\{p^2 \chi^2(\mu \parallel \pi)^{1/p}\}, \qquad \forall \mu \ll \pi \tag{6}$$

with constant $C = 2C_P$. In fact, the converse also holds: the validity of (6) implies the Poincaré inequality (P) with constant $C_P = C/\sqrt{2}$.

If we specialize this result to the case $p = 2$, then the Poincaré inequality (P) implies

$$W_2^2(\mu, \nu) \le 8C_P \sqrt{\chi^2(\mu \,\|\, \pi)}, \qquad \forall \mu \ll \pi. \tag{7}$$

In the next section, we give a proof of the inequality (7) with a slightly worse constant, i.e., with 9 instead of 8.

We now briefly describe the method of [OV00], since it is relevant for our approach. Otto and Villani work in the framework of *Otto calculus*, which interprets LD as the gradient flow of the KL divergence in the space of probability measures equipped with the $W_2$ metric. As discussed in Section 3.2, an LSI is a PL inequality, which ensures rapid convergence of the gradient flow. This is then used to deduce the transportation-cost inequality (4).

We follow the argument of Otto and Villani, but consider the *gradient flow of the chi-squared divergence* instead of the KL divergence. We prove a Łojasiewicz inequality for the chi-squared divergence, and use the gradient flow to deduce (7) (with a slightly worse constant).

## B.2 Proof of the chi-squared transportation inequality

Following the proof outline above, we start by proving a PL-type inequality for the chi-squared divergence. Using tools developed in [AGS08], it is a standard exercise to establish that the Wasserstein gradient of the functional $\mu \mapsto \chi^2(\mu \,\|\, \pi)$ is given by $2\nabla(\mathrm{d}\mu/\mathrm{d}\pi)$. Therefore, the right-hand side of the following inequality involves the squared norm of the Wasserstein gradient of the chi-squared divergence, where we use the norm corresponding to the Riemannian structure of Wasserstein space (see [AGS08, §8]). Note that since the objective is raised to the power $3/2$ on the left-hand side it is not quite a PL inequality, and rather it is a form commonly referred to as a Łojasiewicz inequality [Loj63] with parameter $3/4$.

**Proposition 1.** *Let $C_P \in (0, \infty]$ denote the Poincaré constant of $\pi$. Then,*

$$\chi^2(\mu \,\|\, \pi)^{3/2} \le \frac{9C_P}{4} \int \big\|\nabla \frac{\mathrm{d}\mu}{\mathrm{d}\pi}\big\|^2 \,\mathrm{d}\mu, \qquad \forall \mu \ll \pi.$$

*Proof.* Using the Poincaré inequality (P), we obtain

$$\int \big\|\nabla \frac{\mathrm{d}\mu}{\mathrm{d}\pi}\big\|^2 \,\mathrm{d}\mu = \int \big\|\nabla \frac{\mathrm{d}\mu}{\mathrm{d}\pi}\big\|^2 \frac{\mathrm{d}\mu}{\mathrm{d}\pi} \,\mathrm{d}\pi = \frac{4}{9} \int \big\|\nabla\big(\frac{\mathrm{d}\mu}{\mathrm{d}\pi}\big)^{3/2}\big\|^2 \,\mathrm{d}\pi \ge \frac{4}{9C_P} \mathrm{var}_\pi\big(\big(\frac{\mathrm{d}\mu}{\mathrm{d}\pi}\big)^{3/2}\big).$$

In the following steps, we apply the following: (1) $\mathrm{var}\, X \le \mathbb{E}[|X - c|^2]$ for any $c \in \mathbb{R}$; (2) $x \mapsto x^{2/3}$ is $2/3$-Hölder continuous with unit constant; (3) Jensen's inequality.

$$
\begin{aligned}
\chi^2(\mu \,\|\, \pi) = \mathrm{var}_\pi\big(\frac{\mathrm{d}\mu}{\mathrm{d}\pi}\big) &\stackrel{(1)}{\le} \mathbb{E}_\pi\Big[\big|\frac{\mathrm{d}\mu}{\mathrm{d}\pi} - \mathbb{E}_\pi\big[\big(\frac{\mathrm{d}\mu}{\mathrm{d}\pi}\big)^{3/2}\big]^{2/3}\big|^2\Big] \\
&\stackrel{(2)}{\le} \mathbb{E}_\pi\Big[\big|\big(\frac{\mathrm{d}\mu}{\mathrm{d}\pi}\big)^{3/2} - \mathbb{E}_\pi\big[\big(\frac{\mathrm{d}\mu}{\mathrm{d}\pi}\big)^{3/2}\big]\big|^{4/3}\Big] \\
&\stackrel{(3)}{\le} \mathbb{E}_\pi\Big[\big|\big(\frac{\mathrm{d}\mu}{\mathrm{d}\pi}\big)^{3/2} - \mathbb{E}_\pi\big[\big(\frac{\mathrm{d}\mu}{\mathrm{d}\pi}\big)^{3/2}\big]\big|^2\Big]^{2/3} = \Big(\mathrm{var}_\pi\big(\big(\frac{\mathrm{d}\mu}{\mathrm{d}\pi}\big)^{3/2}\big)\Big)^{2/3}.
\end{aligned}
$$

This proves the result. $\qquad\square$

**Theorem 2.** *Suppose $\chi^2(\cdot \,\|\, \pi)$ satisfies the following Łojasiewicz inequality:*

$$\chi^2(\mu \,\|\, \pi)^{2/q} \le 4C_{PL}\, \mathbb{E}_\mu\big[\big\|\nabla \frac{\mathrm{d}\mu}{\mathrm{d}\pi}\big\|^2\big], \qquad \forall \mu \ll \pi, \tag{8}$$

*for some $q \in (1, \infty)$. Then, $\pi$ satisfies the chi-squared transportation inequality*

$$W_2^2(\mu, \pi) \le p^2 C_{PL}\, \chi^2(\mu \,\|\, \pi)^{2/p}, \qquad \forall \mu \ll \pi,$$

*where $1/p + 1/q = 1$.*

*Proof.* The proof follows [OV00]. Take a path $(\mu_t)_{t \geq 0}$ starting at some $\mu_0 = \mu$ and following the $W_2$ gradient flow of the chi-squared divergence $\chi^2(\cdot \parallel \pi)$, that is,

$$\partial_t \mu_t = 2 \operatorname{div}\left(\mu_t \nabla \frac{\mu_t}{\pi}\right).$$

The existence of this gradient flow and the regularity required for the following computations can be justified by [OT11; OT13] and [AGS08, Theorem 11.2.1]. Denote by $T_t$ the optimal transport map sending $\mu_t$ to $\mu_0$. Then, the time derivative of the squared Wasserstein distance can be computed as in [AGS08, Corollary 10.2.7] to be

$$\partial_t W_2^2(\mu_0, \mu_t) = -4\, \mathbb{E}_{\mu_t}\big\langle \nabla \tfrac{\mu_t}{\pi}, T_t - \mathrm{id}\big\rangle \leq 4 W_2(\mu_0, \mu_t)\, \mathbb{E}_{\mu_t}\big\|\nabla \tfrac{\mu_t}{\pi}\big\|,$$

where we apply the Cauchy-Schwarz and Jensen inequalities. It yields

$$\partial_t W_2(\mu_0, \mu_t) \leq 2\, \mathbb{E}_{\mu_t}\big\|\nabla \tfrac{\mu_t}{\pi}\big\|.$$

Also, the chi-squared divergence satisfies

$$\partial_t \chi^2(\mu_t \parallel \pi) = -4\, \mathbb{E}_{\mu_t}\big[\big\|\nabla \tfrac{\mu_t}{\pi}\big\|^2\big].$$

Using the assumption (8),

$$\partial_t[\chi^2(\mu_t \parallel \pi)^{1/p}] = \frac{\partial_t \chi^2(\mu_t \parallel \pi)}{p\chi^2(\mu_t \parallel \pi)^{1/q}} = -\frac{4}{p\chi^2(\mu_t \parallel \pi)^{1/q}}\, \mathbb{E}_{\mu_t}\big[\big\|\nabla \tfrac{\mu_t}{\pi}\big\|^2\big] \leq -\frac{2}{p\sqrt{C_{\mathsf{PL}}}}\, \mathbb{E}_{\mu_t}\big\|\nabla \tfrac{\mu_t}{\pi}\big\|.$$

If we define

$$g(t) := W_2(\mu_0, \mu_t) + p\sqrt{C_{\mathsf{PL}}}\, \chi^2(\mu_t \parallel \pi)^{1/p},$$

we have proved that

$$g' \leq 0.$$

Since $g(0) = p\sqrt{C_{\mathsf{PL}}}\, \chi^2(\mu_0 \parallel \pi)^{1/p}$ and $\lim_{t \to \infty} g(t) = W_2(\mu, \pi)$, we have shown a transport inequality

$$W_2^2(\mu, \pi) \leq p^2 C_{\mathsf{PL}}\, \chi^2(\mu \parallel \pi)^{2/p}. \qquad \square$$

**Theorem 3.** *Let $\pi$ be a distribution on $\mathbb{R}^d$ with finite Poincaré constant $C_{\mathsf{P}} > 0$. Then for any measure $\mu \in \mathcal{P}_2(\mathbb{R}^d)$, it holds*

$$W_2^2(\mu, \pi) \leq 9 C_{\mathsf{P}} \sqrt{\chi^2(\mu \parallel \pi)}.$$

*Proof.* The inequality follows immediately from the two preceding results. $\qquad \square$

*Remark* 1. Transportation-cost inequalities for Rényi divergences were also studied in [Din14; BD15].

## C   Additional choices for the mirror map

We extend our results to other choices of the mirror map $\phi$ that serve as proxies for $V$ and that also lead to exponential convergence of MLD.

The first result below is useful in situations when there exists a strictly convex mirror map $\phi$ such $\nabla \phi$ is easier to invert than $\nabla V$. It ensures exponential ergodicity of (MLD) when $\nabla^2 V$ dominates $\nabla^2 \phi$ in the sense of the Loewner order.

**Corollary 5.** *Suppose that $\pi$ is strictly log-concave and that $\nabla^2 \phi \preceq C \nabla^2 V$, where $\preceq$ denotes the Loewner order. Then, the law $(\mu_t)_{t \geq 0}$ of MLD satisfies*

$$2\|\mu_t - \pi\|_{\mathrm{TV}}^2,\ H^2(\mu_t, \pi),\ D_{\mathrm{KL}}(\mu_t \parallel \pi),\ \chi^2(\mu_t \parallel \mu),\ \frac{1}{2C_{\mathsf{P}}} W_2^2(\mu_t, \pi) \leq e^{-\frac{2t}{C}} \chi^2(\mu_0 \parallel \pi).$$

*Proof.* The assumption implies

$$C\, \mathbb{E}_\pi\langle \nabla f, (\nabla^2 \phi)^{-1} \nabla f\rangle \geq \mathbb{E}_\pi\langle \nabla f, (\nabla^2 V)^{-1} \nabla f\rangle \geq \mathrm{var}_\pi f,$$

where again we apply the Brascamp-Lieb inequality. This verifies (MP) with constant $C_{\mathsf{MP}} = C$. $\quad \square$

Our second result does not require $\pi$ to be log-concave but only that it is close to a strictly log-concave distribution $\widetilde{\pi}$ in the following sense: the density of $\pi$ with respect to $\widetilde{\pi}$ is uniformly bounded away from $0$ and $\infty$.

**Corollary 6.** *Suppose that $\widetilde{\pi} = \exp(-\widetilde{V})$ is strictly log-concave and suppose that $\pi$ has density $\rho$ w.r.t. $\widetilde{\pi}$. Let $M := (\sup \rho)/(\inf \rho)$. Then, the law $(\mu_t)_{t \geq 0}$ of* MLD *with mirror map $\phi = \widetilde{V}$ and target density $\pi$ satisfies*

$$2\|\mu_t - \pi\|_{\mathrm{TV}}^2, \; H^2(\mu_t, \pi), \; D_{\mathrm{KL}}(\mu_t \| \pi), \; \chi^2(\mu_t \| \mu), \; \frac{1}{2C_{\mathsf{P}}M} W_2^2(\mu_t, \pi) \leq e^{-\frac{2t}{M}} \chi^2(\mu_0 \| \pi),$$

*where $C_{\mathsf{P}}$ is the Poincaré constant of $\widetilde{\pi}$.*

*Proof.* It is standard that the Poincaré inequality (P), and the mirror Poincaré inequality (MP), are stable under bounded perturbations of the measure. It implies that $\pi$ satisfies a Poincaré inequality with constant $C_{\mathsf{P}}M$, and a mirror Poincaré inequality with constant $M$. We prove the latter statement for completeness; for the former statement, see [Han16, Problem 3.20].

Observe that

$$\int \langle \nabla f, (\nabla^2 \widetilde{V})^{-1} \nabla f \rangle \, \mathrm{d}\pi = \int \langle \nabla f, (\nabla^2 \widetilde{V})^{-1} \nabla f \rangle \frac{\mathrm{d}\pi}{\mathrm{d}\widetilde{\pi}} \, \mathrm{d}\widetilde{\pi} \geq (\inf \rho) \int \langle \nabla f, (\nabla^2 \widetilde{V})^{-1} \nabla f \rangle \, \mathrm{d}\widetilde{\pi}$$

and

$$\mathrm{var}_{\widetilde{\pi}} f = \inf_{m \in \mathbb{R}^d} \int \|f - m\|^2 \, \mathrm{d}\widetilde{\pi} = \inf_{m \in \mathbb{R}^d} \int \|f - m\|^2 \frac{\mathrm{d}\widetilde{\pi}}{\mathrm{d}\pi} \, \mathrm{d}\pi$$

$$\geq \frac{1}{\sup \rho} \inf_{m \in \mathbb{R}^d} \int \|f - m\|^2 \, \mathrm{d}\pi = \frac{1}{\sup \rho} \mathrm{var}_\pi f.$$

Combining these inqualities with the Brascamp-Lieb inequality for $\widetilde{\pi}$,

$$\int \langle \nabla f, (\nabla^2 \widetilde{V})^{-1} \nabla f \rangle \, \mathrm{d}\widetilde{\pi} \geq \mathrm{var}_{\widetilde{\pi}} f,$$

yields (MP) with constant $C_{\mathsf{MP}} = M$. $\qquad\square$

# D   Stability in KL with respect to exp-concave perturbations

The following lemma quantifies the approximation error of replacing $\pi$ by $\pi_\beta$ in Section 4.2 and, more generally provides a simple bound to control the KL divergence between a log-concave distribution and its perturbation by a $\nu$-exp-concave barrier function. Its proof uses crucially displacement convexity of the KL divergence to a log-concave measure [Vil03, §5], and it can be viewed as the sampling analogue of [Nes04, (4.2.17)].

Recall that $b$ is $\nu$-*exp-concave* if the mapping $\exp(-\nu^{-1}b)$ is concave.

**Lemma 1.** *Let $\pi$ be a log-concave distribution on a convex set $\mathcal{K} \subset \mathbb{R}^d$. Fix $\nu > 0$, and let $\widetilde{\pi}$ have density $\exp(-b)$ with respect to $\pi$, where $b : \mathcal{K} \to \mathbb{R}$ is $\nu$-exp-concave. Then it holds that*

$$D_{\mathrm{KL}}(\widetilde{\pi} \| \pi) \leq \nu.$$

*Proof.* On $\mathrm{int}\,\mathcal{K}$, we have

$$-\nabla \ln \frac{\mathrm{d}\widetilde{\pi}}{\mathrm{d}\pi} = \nabla b. \tag{9}$$

The measure $\pi$ is log-concave, so by displacement convexity of entropy [AGS08, Theorem 9.4.11] and the "above-tangent" formulation of convexity [Vil03, Proposition 5.29], we have

$$0 = D_{\mathrm{KL}}(\pi \| \pi) \geq D_{\mathrm{KL}}(\widetilde{\pi} \| \pi) + \mathbb{E}\big\langle \nabla \ln \frac{\mathrm{d}\widetilde{\pi}}{\mathrm{d}\pi}(\widetilde{X}), X - \widetilde{X} \big\rangle,$$

where $(X, \widetilde{X})$ are optimally coupled for $\pi$ and $\widetilde{\pi}$. If we rearrange this inequality and use the identities in (9), we get

$$D_{\mathrm{KL}}(\widetilde{\pi} \| \pi) \leq -\mathbb{E}\big\langle \nabla \ln \frac{\mathrm{d}\widetilde{\pi}}{\mathrm{d}\pi}(\widetilde{X}), X - \widetilde{X} \big\rangle = \mathbb{E}\langle \nabla b(\widetilde{X}), X - \widetilde{X} \rangle. \tag{10}$$

We now use the fact that $b$ is $\nu$-exp-concave. To that end, define the convex function

$$\varphi(t) = -\exp\bigl(-\frac{1}{\nu}b(\widetilde{X} + t\,(X - \widetilde{X}))\bigr), \qquad t \in [0,1]\,.$$

By convexity, we have

$$\varphi'(0) \cdot (1 - 0) \leq \varphi(1) - \varphi(0) \leq -\varphi(0) = \exp\bigl(-\frac{1}{\nu}b(\widetilde{X})\bigr).$$

Since

$$\varphi'(0) = \frac{1}{\nu}\exp\bigl(-\frac{1}{\nu}b(\widetilde{X})\bigr)\,\langle\nabla b(\widetilde{X}), X - \widetilde{X}\rangle\,,$$

the above inequality reads $\langle\nabla b(\widetilde{X}), X - \widetilde{X}\rangle \leq \nu$, which completes the proof together with (10). □

*Remark* 2. It is known that given any convex body $\mathcal{C} \subset \mathbb{R}^d$, there exists a standard self-concordant $\nu^{-1}$-exp-concave barrier with $\nu \leq d$ [NN94; BE15; TY18].

# E   Numerical experiments

In this section, we gather additional details and figures to support our numerical experiments. First, in Section E.1, we display the samples from our Gaussian experiment. Then, Section E.2 gives details of the Bayesian logistic regression experiment displayed in Figure 1 and shows the effect of varying step size. Section E.3 gives details of sampling from an ill-conditioned convex set. Finally, Section E.4 shows an experiment where we use the NLA and a Mirror-Langevin Algorithm MLA to approximately sample from a degenerate log-concave distribution.

## E.1   Sampling from a Gaussian distribution

We display some supplementary experiments for the elliptically symmetric scatter family example of Section 5. First, we repeat the example in Figure 4 for the simpler case of the Gaussian distribution ($\gamma = 1$) on $\mathbb{R}^{100}$ with the same scatter matrix $\Sigma = \mathrm{diag}(1, 2, \ldots, 100)$ in Figure 5. We again see the superiority of NLA over the Unadjusted Langevin Algorithm (ULA) [DM+19] and the Tamed Unadjusted Langevin Algorithm (TULA) [Bro+19]. Here and in Section 5 the additional parameter of TULA (denoted $\gamma$ in [Bro+19]) is chosen equal to .1.

Figure 5: We display convergence of the various algorithms for an ill-conditioned Gaussian distribution, with $d = 100$ and $\Sigma = \mathrm{diag}(1, 2, \ldots, 100)$. Left: error is the squared distance from 0. Right: error is the relative distance between scatter matrices. As in the experiment displayed in Figure 4, NLA rapidly converges both in terms of location and scale for large step sizes.

We also display some samples from the Gaussian experiment of Figure 5 in Figure 6. NLA maintains good performance for a wide range of step-size choices, while ULA and TULA require a small step size to accurately sample from the target distribution. In fact, even with a small step size, ULA and TULA often jump to small probability regions, while NLA avoids these regions even for large step sizes.

Figure 6: Samples from NLA, ULA, and TULA for the ill-conditioned Gaussian example of Figure 5, with $\Sigma = \text{diag}(1, 2, \ldots, 100)$. We display the projection onto the first (least spread) and last (most spread) population principal components, along with the projection of a 95% confidence region. Top: the step size for all algorithms is $h = 0.7$, Bottom: the step size for all algorithms is $h = 0.05$.

### E.2 Bayesian logistic regression

We give details for the two-dimensional Bayesian logistic regression example in Figure 1. In the Bayesian logistic regression model, covariates are drawn as $X_i \sim \mathcal{N}(0, \text{diag}(10, 0.1))$, the response variables are $Y_i \sim \text{Ber}(\text{logit}(\theta^\top X_i))$, and the parameters $\theta$ have a $\mathcal{N}(0, 10 I_2)$ prior. We consider using NLA to sample from the posterior distribution of $\theta$ given the observations $(X_i, Y_i), i = 1, \ldots, n$, which is

$$\pi(\theta) \propto \exp\left[ -\frac{1}{20}\|\theta\|^2 + \sum_{i=1}^n Y_i \theta^\top X_i - \ln(1 + e^{\theta^\top X_i}) \right],$$

which is strongly log-concave. While the gradient of the potential is invertible, it has no closed-form, and so in our experiments we invert it numerically by solving $\nabla V^\star(y) = \text{argmax}_{x \in \mathbb{R}^d} \{\langle x, y \rangle - V(x)\}$ with Newton's method. We find that, with a warm start from the current iterate $X_t$, it suffices to run Newton's method for a small number of iterations to approximately invert the gradient.

For the purposes of this experiment, we generate 100 samples $X_i \sim \mathcal{N}(0, \text{diag}(10, 0.1))$ and $Y_i \sim \text{Ber}(\text{logit}(\theta^{\star\top} X_i))$, where we set $\theta^\star = (1, 1)$.

We display the result for various sampling algorithms in Figure 1. All algorithms are implemented with $h = 0.1$ and a burn-in time of $10^4$ steps. This example shows the advantage of taking a large step-size with NLA in this ill-conditioned model, while ULA and TULA create samples that are overdispersed. In Figure 7, we also show the effect of decreasing step size in this example. In this case, we see that ULA and TULA still step into low probability regions or fail to explore the underlying density well. On the other hand, NLA remains constrained in the high probability region.

### E.3 Uniform sampling on a convex body

This section contains details for the simulations in Figure 3. We sample from the uniform distribution on the rectangle $[-0.01, 0.01] \times [-1, 1]$ using NLA, PLA, and the Metropolis-Adjusted Langevin Algorithm (MALA) [Dwi+19]. PLA and MALA target the uniform distribution directly. NLA samples from an approximate distribution, given in Section 4.2. The step sizes are chosen as $h = 10^{-5}$ for NLA and PLA and $h = 0.01$ for MALA. The step sizes for PLA and MALA are tuned to allow the

Figure 7: Samples from the posterior distribution of a Bayesian logistic regression model using one run of NLA, ULA, and TULA after a burn-in of $10^4$. Left: large step size (all algorithms use $h = 0.05$); NLA remains within the high-density contours, while the ULA and TULA take steps into low-density areas. Right: small step size (all algorithms use $h = 0.01$); NLA explores the underlying distribution faster than its competitors.

Figure 8: $W_2$ distance (on logarithmic scale) between the uniform distribution on the rectangle $[-0.01, 0.01] \times [-1, 1]$, and samples produced by NLA, PLA, and MALA.

algorithm to reach approximate stationarity in the fewest number of iterations. MALA can use a larger step size because it is unbiased (its stationary distribution coincides with the target distribution, due to the Metropolis-Hastings adjustment). On the other hand, samples from PLA tend to cluster around the boundary for larger step sizes, so we use a smaller step size for both PLA (and NLA for fair comparison).

To evaluate the performance of the algorithms, we estimate the 2-Wasserstein distance between the samples drawn by the algorithms and samples drawn from the uniform distribution on the rectangle; see Figure 8. We use the Sinkhorn distance ($\varepsilon = 0.01$) as an approximation for the 2-Wasserstein distance [Cut13; AWR17]. Specifically, we sample 1000 points in parallel, using the three algorithms of interest. At each iteration, we also draw 1000 points from the uniform distribution on the rectangle, and we compute the Sinkhorn distance between these points and the samples produced by the algorithms. The convergence estimates are averaged over 30 runs.

### E.4 Approximate sampling from degenerate log-concave distributions

In this section, we explore further the problem of approximately sampling according to the measure $\pi(x) \propto \exp(-\|x\|)$ in $\mathbb{R}^2$ considered in Figure 2. To that end, we use the penalization strategy outlined in Section 4.1 and sample instead from the strongly log-concave measure $\pi_\beta(x) \propto \exp(-\|x\| - \beta\|x - \mathbf{1}\|^2)$ as in Corollary 2, where $\beta = 0.0005$, using discretizations of either NLD or MLD with a customized mirror map. Here, $\mathbf{1}$ is the vector of all ones, which simulates the effect of not knowing the true mean.

We initialize all algorithms with a random point $X_0$ with $\|X_0\| = 1000$. The initialization is intentionally chosen so that the gradients of the potential at initialization are extremely small. In these circumstances, we expect ULA to mix slowly.

Through this experiment, we demonstrate two empirical observations:

1. Initially, the iterates of NLA converge extremely rapidly to the vicinity of the origin. This suggests that NLA can be useful for initializing other sampling algorithms in highly ill-conditioned settings.

2. However, once the iterates of NLA are near the origin, NLA becomes unstable. Specifically, since the Hessian of the potential degenerates rapidly near 0, the iterates of NLA occasionally make large jumps away from 0. This is due to the fact that the Hessian of $V(x) = \|x\| + \beta\|x - \mathbf{1}\|^2$ is given by

$$\nabla^2 V(x) = \frac{1}{\|x\|}\Big[I_2 - \Big(\frac{x}{\|x\|}\Big)\Big(\frac{x}{\|x\|}\Big)^\top\Big] + 2\beta I_2 \tag{11}$$

which blows up to infinity around $x = 0$. We remark that Newton's method in optimization can also exhibit unstable behavior [CGT00; NP06], so this phenomenon is not unexpected. To rectify this behavior, we also consider the Euler discretization of MLD, which we call MLA (see below). We demonstrate that with an appropriate choice of mirror map, the iterates of MLA are stable, yet still enjoy faster convergence than ULA.

Now we proceed to the details of the experiment. We compare four different methods for sampling from this distribution: NLA, ULA, TULA, and the mirror-Langevin Algorithm (MLA)

$$\nabla\phi(X_{k+1}) = \nabla\phi(X_k) - h\nabla V(X_k) + \sqrt{2h}\,[\nabla^2\phi(X_k)]^{1/2}\xi_k, \tag{MLA}$$

with mirror map $\phi(x) = \|x\|^{3/2}$ and potential $V(x) = \|x\| + \beta\|x - \mathbf{1}\|^2$. Notice that this mirror map corresponds to that used in the generalized Gaussian case of Section 5.

In Figure 9, we display the results of the first 1000 iterations of the four algorithms. In this stage of the experiment, we observe rapid convergence of NLA towards the origin (around which the mass is concentrated), and MLA also exhibits faster convergence than ULA and TULA. However, already in Figure 9 (Right) we observe the instability of NLA witnessed through large jumps of the iterates.

Next, in Figure 10, we treat the samples from the first 1000 iterations as burn-in, and we look at the performance of the next 1000 samples. Here we see that the flexible framework of the more general MLD allows us to design algorithms which can outperform NLA with superior stability in specific scenarios.

Recall that the Hessian of the potential $V$ is given in (11) while the potential of the mirror map $\phi$ is given by

$$\nabla^2\phi(x) = \frac{3}{2\|x\|^{1/2}}\Big[I_2 - \frac{3}{4}\Big(\frac{x}{\|x\|}\Big)\Big(\frac{x}{\|x\|}\Big)^\top\Big].$$

From these expressions, it can be checked that Corollary 5 holds with $C \leq 3/(4\sqrt{2\beta})$. On the other hand, the measure $\pi_\beta$ satisfies a Poincaré inequality (P) with constant $C_P \leq 1/(2\beta)$. Heuristically, we therefore expect the mixing time of ULA to scale as $O(\beta^{-1})$, and the mixing time of MLA to scale as $O(\beta^{-1/2})$, which provides an explanation for the rates of convergence observed in Figure 9. In comparison, the mixing time of NLA is scale-invariant, i.e. $O(1)$, as we demonstrated in Corollary 1, as witnessed by the initial rapid convergence in Figure 9.

As mentioned in our open questions, this points to the intriguing possibility of developing more stable variants of NLA, which would mirror the development of such strategies for Newton's method [CGT00; NP06].

Figure 9: First stage of the experiment. Left: We plot the norm of the running mean versus the iteration number for the target measure $\pi_\beta(x) \propto \exp(-\|x\| - 0.0005\|x - \mathbf{1}\|^2)$. Right: We display the corresponding samples.

Figure 10: Second stage of the experiment. In this stage, we treat the 1000 samples from the first stage of the experiment as burn-in and look at the performance of the next 1000 samples. Left: We plot the logarithm of the norm of the running mean versus iteration. Right: We again display the corresponding samples.

## F    Broader impact

The sampling algorithms designed in this paper have the potential to improve a wide variety of Bayesian methods and therefore have an indirect impact on various domains such as health and medicine where such methods are pervasive. Sampling algorithms are also used for the generation of automated spam messages, which have potentially negative effects on society. Since this paper is primarily focused on theory, these questions are not addressed here.

## G    Acknowledgments

Philippe Rigollet was supported by NSF awards IIS-1838071, DMS-1712596, DMS-TRIPODS-1740751. Sinho Chewi and Austin Stromme were supported by the Department of Defense (DoD) through the National Defense Science & Engineering Graduate Fellowship (NDSEG) Program. Thibaut Le Gouic was supported by ONR grant N00014-17-1-2147 and NSF IIS-1838071.

We thank the reviewers for very helpful suggestions regarding the presentation of the paper.

## Footnotes

[1] This ensures that $\nabla\phi$ is invertible, c.f. [Roc97, §26].