[Reviews · NeurIPS 2020]

Review 1

Summary and Contributions: The paper studies the problem of sampling from a multivariate target probability distribution using the state of an ergodic diffusion at time t. The idea is that a careful choice of the drift and the diffusion coefficient ensures that the target distribution is invariant to the action of the semi-group of the corresponding diffusion process. The two important contributions, in my opinion, are the following: 1) A tight rate of contraction of the Langevin diffusion when the target satisfies the Poincaré inequality 2) A proper way of defining the miror version of the Langevin dynamics, along with an elegant result on its contraction rate in terms of the chi-squared divergence.

Strengths: 1. There were several attempts in the ML literature to introduce the mirror version of the Langevin dynamics (LD). However, they did not really succeed in getting better results as compared to the vanilla LD. The proposal made in this paper seems to be the first that really demonstrates the advantage of using the mirror version. 2. All the presented mathematical results are very elegant, with easy-to-follow proofs. 3. The paper is very well polished. The notations are well chosen, which makes the reading a real pleasure. 4. I have rarely seen a conference paper in which the relation to prior work is done in such a thorough way. 5. I like a lot Corollary 4, which establishes contraction under the Poincaré condition. Such a result is conjectured in [Dalalyan, JRSS B 2017]. However, the best result, to my opinion, stated in the prior work requires the log-Sobolev inequality to hold, which is a significantly stronger assumption than requiring merely the Poincaré inequality to hold. 6. There is a good balance between theoretical and experimental results.

Weaknesses: The effect of the discretization is studied only by numerical experiments. This is, however, not an important weakness in my opinion. For a conference paper, I can even say that I approve the choice of focusing only on results in continuous-time.

Correctness: I have checked most mathematical derivations and have found no mistake. I believe that all the results of the paper are correct.

Clarity: One of the clearest papers I have ever reviewed.

Relation to Prior Work: As already mentioned, the discussion of prior work and the differences of the present paper as compared to all the previous results is very well conducted.

Reproducibility: Yes

Additional Feedback: I have just a small comment. When applying the NLD to the quadratic potential V(x) = (1/2)*m*|x|^2, then Y_t = m * X_t and, consequently, the process X is a diffusion process given by the SDE dX_t = -(1/m) * Grad V(X_t) dt + sqrt(2/m) * dB_t. (**) If I am not mistaken, this turns out to be the vanilla LD accelerated by the factor 1/m. From that point of view, it is not quite surprizing that the rate of contraction in Cor 1 has no dependence on the target distribution. The same is true for the time-changed vanilla Langevin process (**) according to corollary 4.


Review 2

Summary and Contributions: This paper provides a nonasymptotic analysis of the mirror-Langevin diffusion in continuous time. Similar to the classical Langevin diffusion, they prove that this diffusion converges to the target exponentially. Moreover, it is shown that this diffusion has a convergence rate independent of the dimension.

Strengths: The analysis is theoretically strong and seems sound (despite I haven't checked every detail).

Weaknesses: Two main weaknesses are that 1) the analysis is conducted just for continuous-time diffusion, but not the practical algorithm, 2) the numerics could have been more comprehensive, especially including some comparisons to the underdamped Langevin schemes (which are known to have better dimension dependence compared to overdamped schemes).

Correctness: Yes.

Clarity: Yes.

Relation to Prior Work: Yes, the discussion of related work seems to be sufficient.

Reproducibility: Yes

Additional Feedback: This paper gives a clean derivation of the exponential convergence of the mirror-Langevin diffusion in continuous-time. This is important since this may then lead to the full analysis of the discrete-time method. I found the overall paper well-written and addressing an important problem. The work is obviously interesting because of a potential connection to its application to an analysis of a discrete scheme. The theory looks clean, sound, and nicely done to obtain exponential convergence this class of diffusions, extending similar results for the classical Langevin diffusion to the mirror-Langevin case. 1) Here the relevant numerical scheme is introduced as the NLA on page 8 while in the conclusion it is said that "Newton schemes" can be unstable in practice. Is the NLA in general a good candidate for this analysis to be useful in practice? Would it be expected that these continuous-time results can be used for the NLA and would it be dimension-free as well? In other words, is the dimension-free convergence rate practically relevant? 2) The experiments are conducted in a setting that could and should be definitely improved. First of all, I am surprised that TULA performs poorly, mostly worse than the ULA. Was it related to the choice of its parameter? Is it the best performance of the TULA that can be obtained? Can authors provide results using different parameter settings of TULA? 3) More on numerics: I feel like the comparison of the performance of the NLA to the methods like HMC or discretized underdamped Langevin diffusion is of great interest here (indeed, more so than ULA and TULA). These methods are known to scale better with dimension, hence any method mentioning dimension dependence as a strength should be compared to these rather than the plain ULA, which is not difficult to beat. I am aware that there are many such proposed methods but comparison to a basic underdamped Langevin scheme would give much more intuition about the scaling w.r.t. dimension compared to second-order schemes. Some small comments: - Figure 2 is not really referred to in the paper - Typo after (NLA) in Section 5, "In cases where \nabla V" should be "nabla^2 V"? Post rebuttal: Authors addressed some of my comments. The dimension dependence of the numerical scheme is not addressed, but I am fine with that, as this requires a lot of work to clarify (possibly in another work).


Review 3

Summary and Contributions: The paper studies the convergence properties of mirror Langevin diffusion (MLD) in continuous-time setting. The convergence analysis is based on selecting Xi-squared divergence as Lyapunov function and assuming a modified version of Poincare inequality called mirror Poincare inequality (MP). Under this assumption, the diffusion process converges exponentially in continuous-time where the convergence rate depends on the MP constant. For the special case of Newton mirror diffusion (NMD), assuming the target distribution is strictly log-concave, the convergence rate is equal to one, independent of dimension and target distribution. The paper also demonstrates the application of MLD for sampling from a uniform distribution on a convex set. Moreover, numerical experiments are given for sampling from a degenerate Gaussian distribution.

Strengths: The paper is nice to read and it is Mathematically rigorous. The way the paper presented the extension of existing ideas to mirror setting was smooth and easy to follow. The connection between the mirror poincare inequality in special Newton setting and the Brascamp-Lieb inequality was nice.

Weaknesses: Although the paper is well-written and nice to read, I found some of the contributions described in the introduction not very surprising. 1. I think the choice of xi-squared for convergence under Poincare inequality is standard. For example, see Proposition 1.3 in [1] (though I am not sure about the original reference). Basically, the L^2 norm is the appropriate norm for analysis of convergence under Poincare inequality (known as variance decay) and Xi-squared is the L^2 norm of density with respect to invariant distribution. The contribution of the paper is extension of this to the mirror Langevin setting. 2. The fact that the convergence rate does not depend on dimension is not surprising: for a strongly convex potential, the Poincare constant is equal to the strong convexity coefficient, independent of dimension. 3. The reason that the strong convexity coefficient does not appear in the convergence rate is an artifact of continuous-time setting. The Newton mirror Langevin can be interpreted as the speed up version of Langevin equation. To be concrete, consider a one-dimensional quadratic potential $V(x) = 1/2 a x^2$. Then, the Langevin eq. is $dX = -aX_t dt + \sqrt{2}dB$ with Poincare constant $a$, and convergence in xi-squared $e^{-at}$. With a time reparametrization $t -> t/a$, one obtains the MLE $dX = -X dt + \sqrt{2/a} dB$ with convergence rate $e^{-t}$. The time reparamtrization is hiding the constant $a$ in the convergence rate of MLE. This is analogous to gradient flow for optimization and the fact that a time reparametrization can artificially speed up convergence. [1] Patrick Cattiauxa and Arnaud Guillin. Trends to equilibrium in total variation distance

Correctness: Yes. To the best of knowledge, they are correct.

Clarity: Yes, It was very nice.

Relation to Prior Work: Yes.

Reproducibility: Yes

Additional Feedback: 1- Adding to the fact that log-Sobolev is stronger than Poincare, it is insightful to note that log-Sobolev implies a sub-Gaussian random variable, while Poincare implies a sub-exponential random variable. 2- It is interesting to understand the properties of the generator of the MLE, compared to LE, since Poincare constant is the spectral gap of the generator. ---------------------after rebuttal-------------------- After reading the response, I change my score to 6. I think it can become a better paper with a little more time and work. - I think it can improve if it contains better discussion of the results: the fact that the Poincare constant does not appear in the rate is because of speed up. I agree with the response that NLD has real effect for non-isotropic Gaussian distribution but that is not captured by the convergence rate result. The Poincare constant is equal to the smallest eigenvalue and independent of the covariance structure. - Also, because it is a theoretical paper, it would be nice to include some characterization of distributions that satisfy the mirror Poincare inequality, similar to the characterizations available for Poincare inequality, like the characterization in [1] obtained with Lyapunov function method. I think this would improve the paper significantly. - And, the numerical experiments can improve if they are more connected to the theoretical results of the paper. I think right now the numerics and theoretical results are not connected. [1] Dominique Bakry, Franck Barthe, Patrick Cattiaux, and Arnaud Guillin. A simple proof of the Poincare ́ inequality for a large class of probability measures including the log-concave case. Electron. Commun. Probab, 13:60–66, 2008.


Review 4

Summary and Contributions: This is mainly a theoretical paper. It proposes a class of diffusions called Newton-Langevin diffusions and proves that they converge to stationarity exponentially fast with a rate which not only is dimension free, but also has no dependence on the target distribution.

Strengths: 1. The theoretical developments are interesting and comprehensive. 2. The empirical examples are relatively simple.

Weaknesses: 1. The assumptions on V such as twice continuously differentiability and strongly convex are not very realistic given the current advances in deep learning. Is it possible to weaken these conditions? 2. In terms of technical tools, can you explicitly describe the novel techniques for proofs beyond Zha+20? 3. Many conclusions are based on the assumptions on some inequalities such as Poincare inequality or PL inequality. Are these smoothness assumptions? There are some conclusions based on conjectures. Can you also make this explicit? 4. Can you discuss the dicretization error? What are the main difficulties for this analysis? 5. The empirical analysis is weak. Is it possible to analyze more challenging distributions? It is also interesting to design an example when V is not convex.

Correctness: Yes

Clarity: Overall, the paper is well written. But there are many abbreviations such as NLA, PLA, NLD, LSI, MP, KLS, .... This makes reading very difficult.

Relation to Prior Work: Except the conclusion, can you emphasize the novelty beyond Zha+20?

Reproducibility: Yes

Additional Feedback:

[Author Response · NeurIPS 2020]

# Main remarks

1. **An analysis focused on the continuous time process**

   Several referees have raised the question of the analysis of the discretized algorithm. We have deliberately avoided this question in order to deliver a crisp and clear message regarding interplay between functional inequalities and the chi-squared divergence.

   The study of discrete-time algorithms brings significant additional details and require additional assumptions that would distract the reader from the main message. In fact, our arXiv preprint has already prompted a successful follow-up study[1] of discretized Langevin using the chi-squared divergence, but again, at the cost of ad-hoc assumptions that may or may not be definitive.

2. **Some rates of convergence are artifacts of continuous time analysis**

   It is true that for an isotropic Gaussian target, NLD is simply a sped-up vanilla LD, but this is no longer the case for other target distributions where NLD has a real effect. In fact, for non-isotropic targets, our experiments demonstrate convincingly that NLD is not just a time-reparametrization of ULA, and that NLD is indeed superior.

3. **There should be comparisons with more algorithms**

   Given the diversity of modifications of ULA, we had decided to add only TULA as a comparator: the improvement of NLA over these algorithms is several orders of magnitude better than the variability within the cluster of ULA modifications. Comparison with Underdamped/Accelerated Langevin (ALA) is presented in the attached figure: it exhibits a behaviour similar to (T)ULA in the anisotropic Gaussian case ($d = 20$). HMC belongs to a different family of algorithms.

# Specific comments

## Reviewer 2

*Instability of the "Newton scheme".* As in any problem in optimization or sampling, we do not advocate for a one-size-fits-all algorithm and, in many examples, additional structure may be leveraged to improve performance. NLA displays generally better behaviour than competitors in this study and, in that sense, is a good off-the-shelf algorithm. Nevertheless, we describe in the appendix an example where the the flexibility of the mirror perspective can be useful to better exploit the structure of the problem.

*TULA vs. ULA.* For the same step size ULA indeed typically outperforms TULA in our experiments but the two have essentially the same behavior. We believe that the range of step sizes in our experiments is sufficient to demonstrate qualitatively the relative behaviour of various algorithms.

## Reviewer 3

*Using the chi-squared divergence is standard.* We completely agree with reviewer on this point and acknowledge inspiration from the Markov semigroup perspective in the text. In fact, for the analysis of vanilla Langevin, the use of the chi-squared divergence is almost a tautology as indicated by our short proof based on semigroups; this fact had somehow been elusive in the sampling literature despite intense activity over the past few years.

*The lack of dimension dependence is not surprising.* For non-strongly-log-concave potentials, whether or not the Poincaré constant depends on the dimension is actually the object of the KLS conjecture (see Sections 3.2 and 4.1). Note that whether this conjecture is true or not, the rate for NLD does not even depend on the Poincaré constant.

## Reviewer 4

*Assumptions.* In the theory of sampling algorithms, we produce some of the weakest conditions amenable to polynomial sampling algorithms.

*Comparison with Zha+20 in terms of technical tools.* In fact, Zha+20 fails to show convergence of the Mirror Langevin Algorithms even for vanishing step size. It also uses some assumptions that are stronger than ours, others that are incomparable, and finally some assumptions that are inherent to their discrete-time analysis.

## Footnotes

[1]M. A. Erdogu and R. Hosseinhazadeh, (2020). A brief note on the convergence of Langevin Monte Carlo in chi-square divergence, arXiv:2007.11612.


[Meta-Review · NeurIPS 2020]

This paper presents a theoretical analyisis of the class of mirror-diffusions in continuous-time; i.e. diffusion processes used to sample from complex target distributions. Whereas these algorithms have been introduced by other authors, the only results available did not show improved convergence rates compared to standard Langevin. Here the authors have obtained improved CV rates under reasonable assumptions (e.g. contraction under the Poincare condition). All the reviewers participated to the discussion after the rebuttal was made available. Strengths: the paper is very well-written, the proofs are clear and easy to follow. Additionally the results are neat and demonstrate the potential usefulness of mirror-diffusions. Weaknesses: the authors have only analyzed the continuous-time algorithm. It is unclear how to discretize efficiently such processes and which theoretical results will be obtained for the discretized version. The authors have only considered Newton-Langevin Algorithm. In most scenarios, for the Newton Langevin Diffusion, the convex conjugate is not available analytically and it is unclear that, when having to solve a convex optimization algorithm at each iteration whether such an algorithm can be practically competitive with underdamped Langevin. The numerics could be improved and better connected to the theoretical part. The authors should also spell out more clearly the limitations of this approach. Overall, despite its weaknesses, this paper presents an interesting theoretical analysis of a new class of diffusions inspired by optimization algorithms. This should motivate further developments in this important area.